# Radiological Biomarkers in MRI directed Rectal Cancer Radiotherapy Volume Delineation

**DOI:** 10.3390/cancers15215176

**Published:** 2023-10-27

**Authors:** Charleen Chan Wah Hak, Svetlana Balyasnikova, Samuel Withey, Diana Tait, Gina Brown, Irene Chong

**Affiliations:** 1The Royal Marsden NHS Foundation Trust, London SW3 6JJ, UK; 2Kingston Hospital NHS Foundation Trust, Kingston upon Thames KT2 7QB, UK; 3Department of Surgery and Cancer, Hammersmith Campus, Imperial College, London W12 0HS, UK

**Keywords:** rectal cancer, radiotherapy, magnetic resonance imaging, target volume delineation, extramural venous invasion, radiological biomarkers

## Abstract

**Simple Summary:**

The correct delineation of disease is a critical step in radiotherapy planning and delivery. In the past, the main focus has been on the primary tumour and any involved lymph nodes. Our study assessed whether the accuracy of gross tumour volume (GTV) contouring for patients with rectal cancer can be improved using an MRI reporting system highlighting areas of contiguous and discontinuous extramural venous invasion (EMVI), known biomarkers associated with poor outcome, as well as the primary tumour and involved nodes. Our study shows that the implementation of an MRI reporting system and detailed radiology discussion improves the accuracy of GTV delineation. This approach can be adopted upfront, at the time of multi-disciplinary team (MDT) discussion, to stratify patients into simple or more complex cases that require increased dedicated radiotherapy planning time and peer review.

**Abstract:**

Our study evaluated whether an MRI reporting system highlighting areas of contiguous and discontinuous extramural venous invasion (EMVI) can improve the accuracy of gross tumour volume (GTV) delineation. Initially, 27 consecutive patients with locally advanced rectal cancer treated between 2012 and 2014 were evaluated. We used an MRI reporting proforma that documented the position of the primary tumour, lymph nodes and EMVI. The new GTVs delineated were compared with historical radiotherapy treatment volumes to identify the frequency of GTV geographical miss. We observed that the delineation of involved nodes and areas of EMVI was more likely to represent sources of uncertainty wherein nodal GTV geographical miss was evident in 5 out of 27 patients (19%). Complete EMVI GTV geographical miss occurred in two patients (7%). We re-evaluated our radiotherapy practice in a further 27 patients after the implementation of a modified MRI reporting system. An improvement was seen; nodal miss was observed in two patients (7%) and partial EMVI miss in one patient (4%), although these areas were encompassed in the planning target volume (PTV). Our study shows that extramural venous invasion and involved nodes need to be highlighted on MRI to improve the accuracy of rectal cancer GTV delineation.

## 1. Introduction

Neoadjuvant chemoradiotherapy (nCRT) improves local control and often results in tumour downstaging for patients with locally advanced rectal cancer [1]. nCRT may lead to significant rates of pathological complete response (pCR) [2,3,4,5,6,7]. Selected patients with clinical and radiological evidence of complete response to nCRT have been managed non-operatively with good oncological outcomes and the advantage of organ preservation [8,9,10,11]. Radiotherapy dose escalation has resulted in increased initial clinical complete responses [12,13]; a higher tumour dose (45 Gy or more) has been identified as an independent factor affecting the frequency of pCR [14]. Furthermore, a number of phase II studies have indicated the presence of a dose–response relationship for tumour regression after nCRT for patients with locally advanced rectal cancer [15,16,17,18], highlighting the importance of accurately boosting the radiotherapy dose for areas of macroscopic disease in selected, high-risk cases.

The delineation of radiotherapy treatment volumes has been shown to be a major source of uncertainty that is extremely likely to impact on treatment quality and clinical outcomes [19]. Whilst intensity-modulated radiation therapy (IMRT) permits a better dose distribution to the target compared with conventional and conformal radiotherapy [20,21], the accuracy of gross tumour volume (GTV) delineation of not only the primary tumour and involved nodes but also areas of extramural venous invasion (EMVI) becomes even more important in the setting of dose intensification [22]. Magnetic resonance imaging (MRI) is considered the gold standard for rectal cancer staging [23,24,25] and accurately defines the depth of invasion through the rectal wall into local structures, extension into the presacral space and mesorectal circumference as well as involved lymph nodes [20]. More recently, high-resolution MRI has also been shown to be important in detecting areas of vascular invasion. EMVI, defined as tumour cells actively invading the veins beyond the muscularis propria, is known to be an independent biomarker of poor prognosis [26,27,28] and leads to an increased risk of disease recurrence for both stage II and stage III rectal cancer, with a detection rate of at least 25% of rectal cancers [29]. In addition, MRI-detected extramural vascular invasion, present in one-third of patients with rectal cancer, is associated with a five-fold increased rate of synchronous metastases and up to a four-fold ongoing risk of developing metastases in follow-up after surgery [30].

Given that the correct delineation of disease is a critical step in the planning and delivery of radiotherapy, the purpose of this study was to assess the accuracy of rectal cancer radiotherapy volume contouring using an MRI reporting proforma that highlights the exact position of the primary tumour, lymph nodes and EMVI compared with historical treatment volumes. We also report a re-evaluation of our radiotherapy practice following this initial study to determine whether specifically drawing attention to involved nodes and areas of contiguous and discontinuous EMVI within formal MRI reporting and at the time of MDT discussion can lead to an improvement in overall disease delineation.

## 2. Materials and Methods

### 2.1. Patient and Tumour Demographics

Twenty-seven consecutive patients with locally advanced rectal cancer treated in the radiotherapy department at the Royal Marsden Hospital between 2012 and 2014 were analysed and retrospectively included in this study. Eligible patients had histologically confirmed high-risk rectal adenocarcinoma defined by the presence of at least one of the following on high-resolution thin-slice MRI (3 mm): tumour within 1 mm of the mesorectal fascia, T3 tumour at or below the level of the levator ani muscle complex, extramural extension ≥ 5 mm, T4 tumour or presence of EMVI. All patients had WHO performance status 0–2. We re-evaluated our practice five years after the initial study with a further 27 patients (second cohort) with locally advanced rectal cancer with discontinuous EMVI treated with chemoradiotherapy at the Royal Marsden Hospital between 2016 and 2020. This second cohort was used to assess whether specifically drawing attention to the location of EMVI in MDT and MRI reporting improved the accuracy of GTV delineation. Permission to include these patients within the context of a service evaluation was obtained from the Royal Marsden Hospital Audit committee.

### 2.2. Chemoradiotherapy Procedure

Each patient received long course chemoradiotherapy with concurrent capecitabine (1650 mg/m^2^/day). Radiation was conformally computer tomography (CT) planned and delivered either with a sequential boost technique (phase 1, 45 Gy in 25 fractions encompassing the primary tumour and pelvic lymph nodes; phase 2, 5.4 Gy in 3 fractions to the assessable tumour with a 2 cm margin in all directions) or using IMRT or RapidArc plans (45 Gy in 25 daily fractions to the elective nodes and 52.5 Gy as a simultaneous integrated boost to the gross tumour).

### 2.3. MRI Proforma

We used an adjusted MRI reporting proforma, verified by gastrointestinal (GI) radiologists specialising in colorectal cancer, that documented the position of EMVI, as well as the original staging MRI pelvis, to delineate gross disease (Figure 1). The same two GI radiation oncologists, who work together as one practice, were involved in this project for the treatment volumes and planning for both the first (2012–2014) and second (2016–2020) patient cohorts.

### 2.4. Target Volume Delineation and Data Collection

Radiotherapy treatment volumes and plans were gathered from the Eclipse and Pinnacle platforms. Radiotherapy volumes were performed as per The Royal Marsden NHS Foundation Trust and Radiation Therapy Oncology Group (RTOG) consensus guidelines [31]. In terms of margins for the primary tumour, clinical target volume (CTV_p_) was GTV_p_ with a 1 cm isotropic margin except anteriorly where 15 mm can be considered for tumours more mobile anteriorly at the discretion of the supervising clinician. For involved lymph nodes, CTV_n_ was GTV_n_ with a 5 mm margin. In the first cohort of patients (2012–2014), PTV was CTV with a 1 cm isotropic margin. However, in light of recent UK guidelines [32], some patients from our second cohort (2016–2020) would have received a smaller PTV margin of CTV with a 5 mm isotropic margin. This emphasises the increasing importance of accurate GTV delineation with the move towards smaller margins. New volumes undertaken using the adjusted MRI reporting proforma were compared with historical radiotherapy treatment volumes. Multiple electronic systems were used to gather patient and tumour demographics, radiological and pathological response as well as resection margin status.

### 2.5. Statistical Analysis

We used a single-stage design whereby if GTV geographical misses are observed in 4 or more out of 27 patients (15%) in this study, then further evaluation of the adjusted MRI reporting proforma is warranted. The threshold was set at 80% power with an alpha of 5% to detect a 20% rate of GTV geographical miss. Weighted kappa coefficient (k) was calculated using Prism online calculator tool (© 2023 GraphPad Software) (https://www.graphpad.com/quickcalcs/kappa1/ (accessed on 3 August 2023)), with the degree of concordance classified as follows: poor, k < 0.4 l; moderate, k = 0.41–0.60; good, k = 0.61–0.75; excellent, k = 0.76–0.80; and almost perfect, k ≥ 0.81.

## 3. Results

### 3.1. Initial Evaluation of Radiotherapy Treatment Volumes Using an Adjusted MRI Reporting Proforma

#### 3.1.1. Patient and Tumour Demographics

Patient and tumour characteristics for the initial cohort are reported in Table 1. All patients in the initial study were diagnosed with histologically proven, high-risk rectal adenocarcinoma treated with CRT between 2012 and 2014. A total of 21 out of 27 patients (78%) were men, and the median age of this first cohort was 69 years (range 54–83 years). Out of 27 patients, 25 (93%) had tumours affecting the mid or low rectum, and 26/27 patients (96%) had T3b-d/T4 disease. Twenty-two patients (81%) had node positive disease, twenty patients (74%) had EMVI present and two patients (7%) presented with metastases (operable liver metastases) (Table 1).

#### 3.1.2. Comparison of Volumes Using Adjusted MRI Reporting with Historical Radiotherapy Volumes

Two GI radiation oncologists and two GI radiologists collaborated to develop an MRI reporting proforma specifically designed to highlight the important aspects of disease depiction that are required to facilitate GTV outlining (Figure 1). Within the proforma, distinct reporting criteria were required; the distance above the anal verge/below the peritoneal reflection, cranio-caudal extension, circumferential position (clock-face) and the presence or absence of a threatened circumferential resection margin (CRM) were recorded for the primary tumour, involved nodes and areas of EMVI. These proformas were then completed by the GI radiologists and used by the radiation oncologists to create new radiotherapy treatment volumes. Through an iterative process, the new treatment volumes were again checked for accuracy by the GI radiologists. The newly derived treatment volumes were then compared with the historical radiotherapy treatment volumes (undertaken by the same GI oncologists) to assess the degree of GTV geographical miss prior to the use of the MRI proforma.

Through treatment volume comparisons, we observed no GTV geographical miss in any of the primary tumours that were all adequately encompassed. However, we observed that delineation of involved lymph nodes and areas of EMVI was more likely to represent sources of uncertainty. Nodal GTV geographical miss in the high-dose boost volume was evident in 5 out of 27 patients (19%); of these, 4 cases (15%) had missed mesorectal nodes that were encompassed in the PTV_45Gy_, but one node involving the pelvic side wall was outside of the treatment volume in the remaining patient (Figure 2A,B). EMVI GTV geographical miss occurred in 2 out of 27 (7%) patients (Figure 3A,B). These areas of EMVI were both encompassed in the PTV_45Gy_. Taken together, these results show proof of concept that an adjustment of MRI reporting to highlight the exact extent and location of macroscopic disease, in particular nodal involvement and EMVI, has the potential to improve the accuracy of radiotherapy volume delineation.

### 3.2. Re-Evaluation following Routine Use of Adjusted MRI Reporting

We considered the observations and experience from our first cohort of patients from 2012 to 2014 in 2015 to identify pragmatic day-to-day modifications within the context of our rectal cancer practice that can lead to more accurate radiotherapy delivery. First, the radiology team implemented a modified reporting system incorporating information that had been shown to be useful from the MRI proforma (Figure 1). Second, as part of the initial patient discussion at the colorectal MDT, the GI radiologists presenting at the meeting were careful to point out the most superior involved nodes as well areas of contiguous and discontinuous EMVI. The radiation oncologists then undertook radiotherapy treatment volume delineation equipped with information from the detailed MRI report and the MDT discussion. We then formally re-evaluated the accuracy of our radiotherapy delineation as described below.

#### 3.2.1. Patient Demographics

Twenty-seven further patients treated between 2016 and 2020 were included in the re-evaluation phase of this study (Table 2). In comparison with patients in the initial study whereby 74% (20 of 27) of patients had tumours with EMVI, 89% (24 of 27) of the patients in the re-evaluation cohort harboured tumours that displayed either contiguous or discontinuous EMVI. These patients were purposefully selected for increased case complexity, as our initial study highlighted that high-risk patients, nodal and EMVI involvement, may benefit most from our proforma implementation. Patient and tumour characteristics for the second cohort are reported in Table 2; 15 out of 27 patients (56%) were men. The median age was 56 years; 26/27 patients (96%) had T3b-d/T4 disease, and 24/27 (89%) had EMVI present. Only one patient (4%) had operable liver metastases at presentation; the remaining patients did not have metastatic disease.

#### 3.2.2. Retrospective Assessment of GTV Delineation by GI Radiologists

The GTVs were retrospectively evaluated by two independent GI radiologists for all 27 patients included in the re-evaluation phase. Areas of GTV geographical miss were recorded. Similar to our initial study, our results revealed that all primary tumours were adequately encompassed within the GTV_p_. Nodal assessment showed that there was geographical miss for two patients (7%) within the GTV_n_, but all involved nodes were encompassed by the PTV_45Gy_. We also reviewed the accuracy of EMVI delineation for each of the 27 cases. Bearing in mind that most of these cases were EMVI positive (89%), an independent radiology review showed that there was partial GTV geographical miss in only one patient (4%) (Figure 4). All things considered, this formal re-evaluation of our practice following adjusted MRI reporting and detailed discussion in MDT has shown a notable improvement in GTV delineation with respect to involved nodes and areas of EMVI.

### 3.3. MRI and Histopathological Response Assessment following CRT

Surgery was performed following CRT in 18/27 (67%) of patients in cohort 1 and 22/27 (81%) of patients in cohort 2. The median time between the completion of radiotherapy and the date of surgery was similar between both groups at 2 months, first cohort (range: 1–9 months) and second cohort (range 1–8 months). The pathological Mandard TRG (pTRG) score on surgical histopathology, where available, showed similar rates of pTRG 1–2 scores between cohorts, 9/17 (53%) patients from the first cohort (2012–2014) and 10/21 (48%) from the second cohort (2016–2020) (Table 3). Radiological TRG grading (rTRG) 1–2 was 8/24 (33%) and was higher from the first cohort compared with 5/25 (20%) from the second cohort (Table 4), which may reflect that higher-risk patients were selected to be included in the second cohort. Overall, the concordance between pTRG and rTRG scores was poor (k = 0.134–0.170), which was consistent with the low agreement (k = 0.24–0.25) between rTRG and pTRG in rectal cancer patients from a previous study [33].

## 4. Discussion

Radiotherapy volume delineation has become more detailed not only with advancements in planning techniques but also with the improved understanding of radiological prognostic biomarkers such as EMVI. As we understand this disease better from a radiological perspective, it has become apparent that the time and effort in tumour delineation can significantly vary according to the complexity of each rectal tumour. It is important to identify these patients as early as possible to facilitate the radiotherapy workflow, thereby maximising the best outcomes.

Our study has shown that the implementation of an adjusted MRI reporting proforma that localises the exact position of the primary tumour, involved lymph nodes and EMVI can improve the accuracy of rectal cancer radiotherapy volume contouring. This proforma represents a systematic appraisal of critical areas to include in the radiotherapy field to help guide GTV delineation and has highlighted the need to focus on the location of involved lymph nodes and EMVI, since there was no GTV geographical miss of the primary tumours.

This study reinforces the value of the MDT forum where MRI scans are reviewed upfront as it is the ideal setting to stratify patients into simple versus complex cases. We suggest that identified complex cases would require this proforma to be used, and preparations can be made for more involved radiotherapy planning time, which may involve further discussions or clarifications between GI radiologists and the treating radiation oncologist. This study is limited as it is a single-centre retrospective evaluation that may not be generalisable to all patient populations or healthcare settings. However, our study represents real-world quality improvement via the audit process.

The next stage would be to trial this approach within a multi-centre prospective evaluation where the impact on longer-term patient outcomes can be assessed. This can be facilitated by prospective recruitment of patients into a study involving adjusted MRI reporting whereby the presence of lymph node and EMVI involvement are identified upfront in the MDT setting. The complex cases identified can thus be flagged for dedicated radiotherapy planning time with radiology input and peer review.

## 5. Conclusions

As our understanding of radiological biomarkers that reflect tumour phenotype, such as EMVI, evolves for patients with rectal cancer, radiotherapy disease delineation will inevitably become more complex. MRI reporting is most relevant in the case of a boost strategy since this is not standard in all countries. Our study shows that the implementation of an MRI reporting system and detailed radiology discussion enhances the accuracy of GTV delineation. This approach can be adopted upfront, at the time of MDT, to stratify patients into simple or more complex cases that require increased dedicated radiotherapy planning time and peer review.

## Figures and Tables

**Figure 1 cancers-15-05176-f001:**
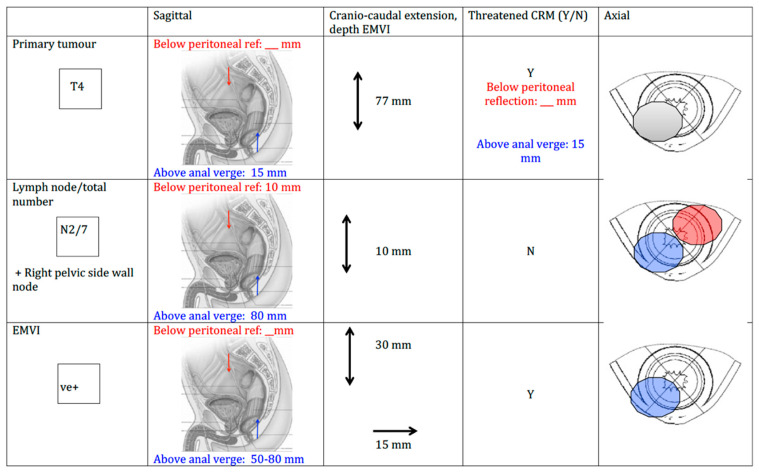
MRI proforma. The distance above the anal verge/below the peritoneal reflection, craniocaudal extension, circumferential position (clock face) and presence or absence/absence of a threatened CRM were recorded for the primary tumour, involved nodes and areas of EMVI.

**Figure 2 cancers-15-05176-f002:**
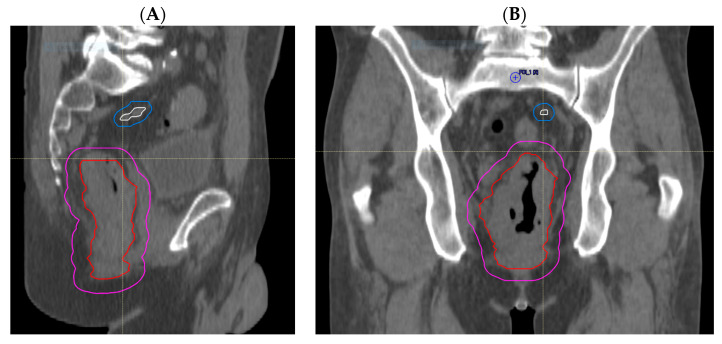
Radiotherapy planning CT showing nodal GTV geographical miss. (**A**) Sagittal view and (**B**) coronal view showing nodal disease (white), with its corrected CTV (blue) extending to the right pelvic sidewall, which is not encompassed by the treated CTV (magenta), with GTV_p_ (red) as shown.

**Figure 3 cancers-15-05176-f003:**
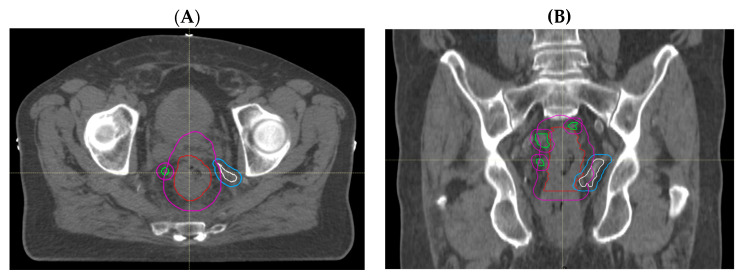
Radiotherapy planning CT showing EMVI GTV geographical miss. (**A**) Sagittal view and (**B**) coronal view illustrating an area of contiguous EMVI (white) with its corrected CTV (blue) extending outside the treated CTV (magenta), with GTV_p_ (red) and GTV_n_ (green) as depicted.

**Figure 4 cancers-15-05176-f004:**
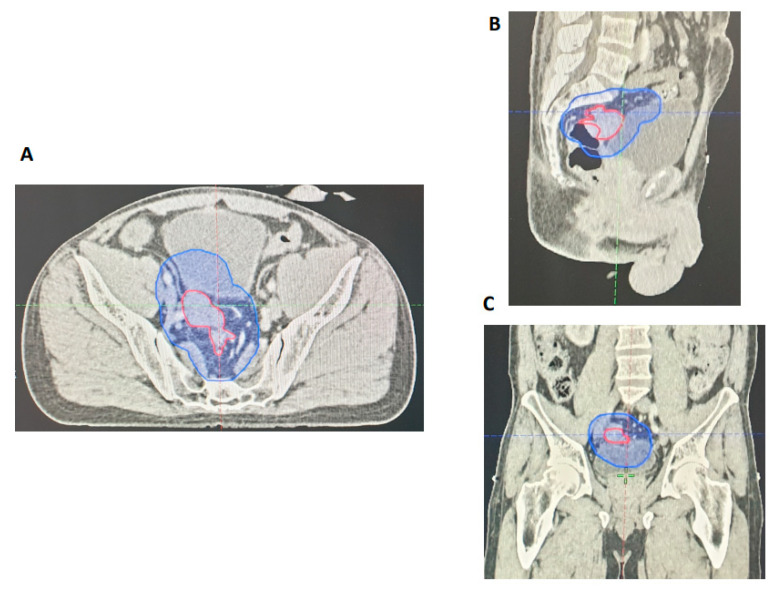
Radiotherapy planning CT showing contiguous EMVI extending from the primary tumour (red) encompassed within the PTV (blue). (**A**) Axial view. (**B**) Sagittal view. (**C**) Coronal view.

**Table 1 cancers-15-05176-t001:** Baseline Patient Demographics and Clinical Characteristics (2012–2014).

Demographic or Clinical Characteristic	All Treated Patients (*n* = 27) N (%)	CRM Involved (*n* = 27) N (%)	CRM Negative (*n* = 5) N (%)
Sex			
Men	21 (78)	18 (82)	3 (60)
Women	6 (22)	4 (18)	2 (40)
Age, years			
Median	69	68.5	69
Range	54–83	54–82	57–83
ECOG Performance Status			
0	13 (48)	11 (50)	2 (40)
1	13 (48)	11 (50)	2 (40)
2	1 (4)	0 (0)	1 (20)
MR TNM Stage			
T2	1 (4)	1 (5)	0 (0)
T3b-d	20 (74)	15 (68)	5 (100)
T4	6 (22)	6 (27)	0 (0)
N0	5 (19)	5 (23)	0 (0)
N1	20 (74)	16 (60)	4 (80)
N2	2 (7)	1 (4)	1 (20)
M0	25 (93)	20 (91)	5 (100)
M1	2 (liver) (7)	2 (liver) (9)	0 (0)
MR EMVI			
Negative	7 (26)	5 (23)	2 (40)
Positive	20 (74)	17 (77)	3 (60)
Location *			
Low	14 (52)	12 (55)	2 (40)
Mid	11 (41)	8 (36)	3 (60)
High	2 (7)	2 (9)	0 (0)
Histopathology (Differentiation Grade)			
Moderate	23 (85)	18 (82)	5 (100)
Poor	2 (7)	2 (9)	0 (0)
Not available	2 (7)	2 (9)	0 (0)
Surgery performed			
Yes	18 (67)	14 (64)	4 (80)
No	9 (33) Died during RT (*n* = 1) Declined surgery (*n* = 2) Deferral of surgery non-intervention arm of trial (*n* = 2) Disease progression (*n* = 1) Unfit for surgery (*n* = 1) CRT delivered for recurrence (*n* = 2)	8 (36) Died during RT (*n* = 1) Declined surgery (*n* = 2) Deferral of surgery non-intervention arm of trial (*n* = 1) Disease progression (*n* = 1) Unfit for surgery (*n* = 1) CRT delivered for recurrence (*n* = 2)	1 (20) Deferral of surgery non-intervention arm of trial (*n* = 1)

* UICC classification used for low (from anal verge up to 6 cm)/mid (6–12 cm)/upper tumours (12–16 cm).

**Table 2 cancers-15-05176-t002:** Baseline Patient Demographics and Clinical Characteristics (2016–2020).

Demographic or Clinical Characteristic	All Treated Patients (*n* = 27) N (%)	CRM Involved (*n* = 23) N (%)	CRM Negative (*n* = 4) N (%)
Sex			
Men	15 (56)	13 (57)	2 (50)
Women	12 (44)	10 (43)	2 (50)
Age, years			
Median	56	54	72.5
Range	33–83	33–83	63–79
ECOG Performance Status			
0	18 (67)	17 (74)	1 (25)
1	8 (30)	5 (22)	3 (75)
2	1 (3)	1 (4)	0 (0)
MR TNM Stage			
T2	1 (4)	1 (5)	0 (0)
T3b-d	17 (63)	13 (57)	4 (100)
T4	9 (33)	9 (39)	0 (0)
N0	2 (7)	2 (9)	0 (0)
N1	25 (93)	21 (91)	4 (100)
N2	0 (0)	0 (0)	0 (0)
M0	26 (96)	22 (96)	4 (100)
M1	1 (liver) (4)	1 (4)	0 (0)
MR EMVI			
Negative	3 (11)	3 (13)	0 (0)
Positive	24 (89)	20 (87)	4 (100)
Location *			
Low	13 (48)	12 (52)	1(25)
Mid	10 (37)	8 (35)	2 (50)
High	4 (15)	3 (13)	1 (25)
Histopathology (Differentiation Grade)			
Moderate	20 (74)	17 (74)	3 (75)
Poor	7 (26)	6 (26)	1 (25)
Surgery Performed			
Yes	22 (81)	18 (78)	4 (100)
No	5 (19) Declined surgery (n = 1) Solitary liver metastasis RR (n = 1) Awaiting completion of adjuvant chemotherapy (n = 1) TRIGGER (deferral of surgery arm) (n = 2)	5 (22) Declined surgery (n = 1) Solitary liver metastases RR (n = 1) Awaiting completion of adjuvant chemotherapy (n = 1) TRIGGER (deferral of surgery arm) (n = 2)	0 (0)

* UICC classification used for low (from anal verge up to 6 cm)/mid (6–12 cm)/upper tumours (12–16 cm).

**Table 3 cancers-15-05176-t003:** Pathological Mandard TRG score on surgical histopathology.

**TRG Grading**	**2012–2014 (*n* = 17)** **N (%)**	**2016–2020 (*n* = 21)** **N (%)**
TRG 1: No residual cancer	5 (29)	2 (10)
TRG 2: Rare residual cancer cells	4 (24)	8 (38)
TRG 3: Fibrosis outgrowing residual cancer	7 (41)	11 (52)
TRG 4: Residual cancer outgrowing fibrosis	1 (6)	0 (0)
TRG 5: Absence of regressive changes	0 (0)	0 (0)

**Table 4 cancers-15-05176-t004:** Radiological TRG score.

**TRG Grading**	**2012–2014 (*n* = 24)** **N (%)**	**2016–2020 (*n* = 25)** **N (%)**
TRG 1: Complete radiological response	1 (4)	3 (12)
TRG 2: Good response	7 (29)	2 (8)
TRG 3: Moderate response	10 (42)	11 (44)
TRG 4: Slight response	5 (21)	9 (36)
TRG 5: No response	1 (4)	0 (0)

## Data Availability

The data presented in this study are available on request from the corresponding author. The data are not publicly available due to privacy restrictions.

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
