# Peer review of "Radiological Biomarkers in MRI directed Rectal Cancer Radiotherapy Volume Delineation"

_cancers, 2023, doi:10.3390/cancers15215176_

Round 1
Reviewer 1 Report
Comments and Suggestions for Authors
- The topic is very interesting; the results and the conclusion are consistent with the aim;
- The manuscript is well written, concise and well organized, with useful and clear tables and good quality of figures; it involves excellent work and interesting observations;
- No changes are needed in the text.
Author Response
We are grateful for the reviewers time and consideration in reviewing our manuscript. We have addressed their comments and suggestions point-by-point in the following letter as shown in red font and the corresponding revisions in track changes in the re-submitted file.
Reviewer 1
The topic is very interesting; the results and the conclusion are consistent with the aim; The manuscript is well written, concise and well organized, with useful and clear tables and good quality of figures; it involves excellent work and interesting observations; No changes are needed in the text.
We would like to thank the reviewer for their time in reviewing our manuscript and their positive feedback. We hope our topic will be of value to our readers.
Reviewer 2 Report
Comments and Suggestions for Authors
The authors present the results of an interesting proof of concept about correct contouring of GTV with implementation of an MRI reporting system in locally advanced colorectal cancer treated with a radiotherapy boost .
Major concerns:
Material and Methods
In principal a geographical miss rarely occurs due to large CTVs. However, the paper lacks any information regarding CTV. E.g. was a specific guideline used?
What were the PTV margins?
The interval between the 2 cohorts is considerate: were there any differences between contouring guidelines and margins apart from the mentioned treatment technique? This could influence the outcome of the comparison.
Although the GTV boost is the main goal I think the questions above need to be answered to get a full overview of the cohorts.
Statistical analysis
The statistical paragraph needs clarification: what are the authors exactly measuring. What is geographical miss? Target outside of PTV? Was there also partial miss?
Were the GI rad oncs the same in both groups?
Results
Where were the (missed) lymph nodes located?
Please clarify line 159: encompassed by GTV or CTV or PTV?
Line 160: LNs and EMVI represents uncertainty. Measured by what? Can you use a metric to express this mathematically?
Discussion
The authors compare two different patient groups with and without MRI reporting system which could result in bias. This should be addressed in the discussion.
Minor concerns
Figures: add CTV to Fig 2 and 3
2.2 Chemoradiotherapy procedure
Suggest to use “ sequential boost” instead of a two phase technique.
Conclusion
Add that the MRI reporting is most relevant in case of a boost strategy, since this is not standard in all countries.
Title:
What does treatment planning stratification refer to? This is not mentioned in the paper. Consider changing the title.
Author Response
We are grateful for the reviewers time and consideration in reviewing our manuscript. We have addressed their comments and suggestions point-by-point in the following letter as shown in red font and the corresponding revisions in track changes in the re-submitted file.
Reviewer 2
The authors present the results of an interesting proof of concept about correct contouring of GTV with implementation of an MRI reporting system in locally advanced colorectal cancer treated with a radiotherapy boost.
We would like to thank the reviewer for taking the time to review our manuscript and providing their insightful comments and suggestions. We have addressed each of their points below and are grateful that their suggested changes have increased the quality of this manuscript and its readability.
Major concerns:
Material and Methods:
In principal a geographical miss rarely occurs due to large CTVs. However, the paper lacks any information regarding CTV. E.g. was a specific guideline used?
Thank you for raising this important point. The Royal Marsden NHS Foundation Trust radiotherapy guidelines and Radiation Therapy Oncology Group (RTOG) consensus guidelines for anorectal cancer were used for contouring of all patients. In terms of margins for the primary tumour, CTVp was GTVp with a 1 cm isotropic margin except anteriorly where 15 mm can be considered for tumours more mobile anteriorly at the discretion of the supervising clinician. For involved lymph nodes, CTVn was GTVn with a 5 mm margin. We have added these details in page 3, lines 119-124.
What were the PTV margins?
The interval between the 2 cohorts is considerate: were there any differences between contouring guidelines and margins apart from the mentioned treatment technique? This could influence the outcome of the comparison.
Although the GTV boost is the main goal I think the questions above need to be answered to get a full overview of the cohorts.
In the first cohort of patients (2012-2014), PTV was CTV with a 1 cm isotropic margin. However, in light of recent UK guidelines from the Royal College of Radiologists, some patients from our second cohort (2016-2020) would have received a smaller PTV margin of CTV with a 5 mm isotropic margin. This emphasises the increasing importance of accurate GTV delineation with the move towards smaller margins. We have added these details in page 3, lines 124-128.
Statistical analysis:
The statistical paragraph needs clarification: what are the authors exactly measuring. What is geographical miss? Target outside of PTV? Was there also partial miss?
Thank you for allowing us to clarify this important point. We defined geographic miss as GTV geographical miss where disease is not encompassed in the GTV. To improve clarity, we have updated this throughout the manuscript where “geographical miss” is mentioned, now reads “GTV geographical miss” (page 1: lines 27, 29, 30; page 3: line 138; page 5: line 206; page 6: lines 213, 216, 220, 229, 235; page 7: lines 348, 354; page 10: line 419).
In relation of GTV geographical miss to the PTV and partial misses, we have detailed for the first patient cohort in page 6, lines 216-221: “Nodal GTV geographical miss in the high dose boost volume was evident in five out of 27 patients (19%); of these, four cases (15%) had missed mesorectal nodes that were encompassed in the PTV45Gy, but one node involving the pelvic side wall was outside of the treatment volume in the remaining patient (Figures 2A and B). EMVI GTV geographical miss occurred in two out of 27 (7%) patients (Figures 3A and B). These areas of EMVI were both encompassed in the PTV45Gy”. For the second cohort we have detailed in page 7, lines 350-355: “Nodal assessment showed that there was geographical miss for two patients (7%) within the GTVn, but all involved nodes were encompassed by the PTV45Gy. (…) independent radiology review showed that there was partial GTV geographical miss in only one patient (4%) (Figure 4)”.
Were the GI rad oncs the same in both groups?
Yes, the same two GI radiation oncologists who work together as one practice were involved in this project for both groups. We have clarified this in our Methods section, page 3, lines 112-115: “The same two GI radiation oncologists, who work together as one practice, were involved in this project for the treatment volumes and planning for both the first (2012-2014) and second (2016-2020) patient cohorts.”
Results:
Where were the (missed) lymph nodes located?
Four of the five missed LNs were mesorectal lymph nodes and one of the lymph nodes missed was involving the pelvic side wall. We have added these details in page 6, lines 216-219, to read: “Nodal GTV geographical miss in the high dose boost volume was evident in five out of 27 patients (19%); of these, four cases (15%) had missed mesorectal nodes that were encompassed in the PTV45Gy, but one node involving the pelvic side wall was outside of the treatment volume in the remaining patient (Figures 2A and B).” We could not conclude on any discrete pattern in location of missed lymph nodes with these small patient numbers.
Please clarify line 159: encompassed by GTV or CTV or PTV?
We have updated this line, page 6, now line 213, “Through treatment volume comparisons, we observed no GTV geographical miss in any of the primary tumours which were all adequately encompassed.”
Line 160: LNs and EMVI represents uncertainty. Measured by what? Can you use a metric to express this mathematically?
Yes, in our study we have stated in page 6, now line 215-216, “that delineation of involved lymph nodes and areas of EMVI were more likely to represent sources of uncertainty”. This was our finding given in both study cohorts, any GTV geographical misses identified only occurred in lymph nodes and EMVI, with no misses of the primary tumour.
Our measured metric was the percentage of cases with GTV geographical miss. In the first cohort, 19% of cases had lymph node misses and 7% of cases had EMVI misses (page 6, lines 217-220) and in the second cohort, 7% of cases had lymph node misses and 4% had EMVI misses (page 7, lines 351-355).
Discussion:
The authors compare two different patient groups with and without MRI reporting system which could result in bias. This should be addressed in the discussion.
Thank you for this comment. The purpose of our study was to evaluate the impact of a structured MRI reporting system. As such, the comparison of two patient groups with and without this intervention was required. However, this was a retrospective study which has its many limitations which we have highlighted in page 10, lines 426-434, “This study is limited as it is a single-centre retrospective evaluation which may not be generalisable to all patient populations or healthcare settings. However, our study represents real-world quality improvement via the audit process. The next stage would be to trial this approach within a multi-centre prospective evaluation where the impact on longer-term patient outcomes can be assessed. This could be facilitated by prospective recruitment of patients into a study where adjusted MRI reporting whereby the presence of lymph node and EMVI involvement are identified upfront in the MDT setting. The complex cases identified can thus be flagged for dedicated radiotherapy planning time with radiology input and peer review.”
Minor concerns
Figures: add CTV to Fig 2 and 3
Thank you, we have updated Figures 2 and 3 and their legends to reflect the CTV.
On page 6, lines 229-231, Figure 2 legend now reads: “Radiotherapy planning CT showing nodal GTV geographical miss. A. Sagittal view and B. Coronal view showing nodal disease (white), with its corrected CTV (blue) extending to the right pelvic sidewall, which is not encompassed by the treated CTV (magenta), with GTVp (red) as shown.”
On page 6, lines 235-237, Figure 3 legend now reads: “Radiotherapy planning CT showing EMVI GTV geographical miss. A. Sagittal view and B. Coronal view illustrating an area of contiguous EMVI (white) with its corrected CTV (blue) extending outside the treated CTV (magenta), with GTVp (red) and GTVn (green) as depicted.”
2.2 Chemoradiotherapy procedure
Suggest to use “sequential boost” instead of a two phase technique.
Thank you, we have replaced “in a two phase technique” with “with a sequential boost technique” in page 3, line 103.
Conclusion:
Add that the MRI reporting is most relevant in case of a boost strategy, since this is not standard in all countries.
Thank you for this comment, we have added this to our conclusion in page 10, lines 438-439, “MRI reporting is most relevant in the case of a boost strategy since this is not standard in all countries.”
Title:
What does treatment planning stratification refer to? This is not mentioned in the paper. Consider changing the title.
We have referred to treatment planning stratification as a decision made at MDT discussion, on review of MRI radiological biomarkers, to classify patients into simple or complex cases. This would allow identification of the morecomplex cases that require increased dedicated radiotherapy planning time and flag them for peer review. However, we take the reviewer’s point that that this is a discussion point of the manuscript rather than in the main body. We have removed “treatment planning stratification” from our title on page 1, lines 2-3: “Radiological Biomarkers in MRI directed Rectal Cancer Radiotherapy Volume Delineation”.
Reviewer 3 Report
Comments and Suggestions for Authors
Dear authors,
Congratulations on your hard work.
Please find below the minor issues that should be corrected in my opinion:
The manuscript acknowledges that it is a single-center retrospective study, which limits the generalizability of the findings. The results may not be representative of broader patient populations or healthcare settings.
The study involved a relatively small sample size, which may affect the statistical power and the ability to draw definitive conclusions. A larger, multi-center study could provide more robust evidence.
In some instances, the manuscript lacks specific details or statistical information, such as the number of patients with EMVI or the specific results of concordance between radiological and pathological assessments.
While the study discusses the potential for future multi-center prospective evaluations, it would be beneficial to provide more concrete suggestions or recommendations for further research to build upon this study's findings.
There are a few formatting and grammar issues throughout the manuscript that could be addressed for clarity and readability.
Comments on the Quality of English Language
There are a few formatting and grammar issues throughout the manuscript that could be addressed for clarity and readability.
Author Response
We are grateful for the reviewers time and consideration in reviewing our manuscript. We have addressed their comments and suggestions point-by-point in the following letter as shown in red font and the corresponding revisions in track changes in the re-submitted file.
Reviewer 3
Dear authors,
Congratulations on your hard work.
Please find below the minor issues that should be corrected in my opinion:
The manuscript acknowledges that it is a single-center retrospective study, which limits the generalizability of the findings. The results may not be representative of broader patient populations or healthcare settings.
The study involved a relatively small sample size, which may affect the statistical power and the ability to draw definitive conclusions. A larger, multi-center study could provide more robust evidence.
We acknowledge this and have expanded on the reviewer’s comment in page 10, lines 426-427: “This study is limited as it is a single-centre retrospective evaluation which may not be generalisable to all patient populations or healthcare settings.”
In some instances, the manuscript lacks specific details or statistical information, such as the number of patients with EMVI or the specific results of concordance between radiological and pathological assessments.
We thank the reviewer for this comment. The numbers of patients from the two cohorts with EMVI are detailed in Table 1 (page 4) and Table 2 (page 8) under “MR EMVI: Negative, Positive”. However, to improve clarity and readability, we have added these numbers in the main text in page 5, lines 190-192: “Twenty-two patients (81%) had node positive disease, twenty patients (74%) had EMVI present and two patients (7%) presented with metastases (operable liver metastases) (Table 1).”; and in page 7, lines 341-343: “The median age was 56 years, 26/27 patients (96%) has T3b-d/T4 disease and 24/27 (89%) had EMVI present.”
The specific results of concordance between radiological (rTRG) and pathological (pTRG) response assessments are detailed in page 7, lines 369-371: “Overall, concordance between pTRG and rTRG scores was poor (k=0.134-0.170), which was consistent with the low agreement (k=0.24-0.25) between rTRG and pTRG in rectal cancer patients from a previous study [33].”
While the study discusses the potential for future multi-center prospective evaluations, it would be beneficial to provide more concrete suggestions or recommendations for further research to build upon this study's findings.
Thank you for this comment. We have expanded on this in page 10, lines 429-434: “The next stage would be to trial this approach within a multi-centre prospective evaluation where the impact on longer-term patient outcomes can be assessed. This could be facilitated by prospective recruitment of patients into a study where adjusted MRI reporting whereby the presence of lymph node and EMVI involvement are identified upfront in the MDT setting. The complex cases identified can thus be flagged for dedicated radiotherapy planning time with radiology input and peer review.”
There are a few formatting and grammar issues throughout the manuscript that could be addressed for clarity and readability.
Thank you, we have proofread our manuscript in detail and hope to have corrected any formatting and grammar issues to improve clarity and readability.
Round 2
Reviewer 2 Report
Comments and Suggestions for Authors
The authors have adequately responded to all raised concerns and changed the manuscript accordingly.
Reviewer 3 Report
Comments and Suggestions for Authors
Dear Authors,
Congratulations! Your revised manuscript version is much more improved. I have no further recommendations on it.